# Evolution Landscape of PiggyBac (PB) Transposon in Beetles (Coleoptera)

**DOI:** 10.3390/genes16121521

**Published:** 2025-12-18

**Authors:** Quan Wang, Shasha Shi, Bingqing Wang, Xin Chen, Naisu Yang, Bo Gao, Chengyi Song

**Affiliations:** College of Animal Science and Technology, Yangzhou University, Yangzhou 225009, China; mx120230886@stu.yzu.edu.cn (Q.W.); dx120220158@stu.yzu.edu.cn (S.S.); mx120220902@stu.yzu.edu.cn (B.W.); dx120240185@stu.yzu.edu.cn (X.C.); naisu.yang@yzu.edu.cn (N.Y.); bgao@yzu.edu.cn (B.G.)

**Keywords:** PiggyBac, transposon, genome, evolution, beetles, insects, arthropods

## Abstract

**Background/Objectives**: The PB family of “cut-and-paste” DNA transposons shows great promise as genetic manipulation tools while significantly impacting eukaryotic genome evolution. However, their evolutionary profile in beetles (Coleoptera), the most species-rich animal order, remains poorly characterized. **Methods**: A local tBLASTN search was conducted to mine PiggyBac (PB) transposons across 136 coleopteran insect genomes, using the DDE domain of the PB transposase as the query. Multiple sequence alignment was performed with MAFFT, and a maximum likelihood phylogenetic tree of the transposase DDE domains was constructed using IQ-TREE. Evolutionary dynamics were analyzed by means of K-divergence. **Results**: Our study reveals PB transposons are widely distributed, highly diverse, and remarkably active across beetles. We detected PB elements in 62 of 136 examined species (45%), classifying them into six distinct clades. A total of 62 PB-containing species harbored intact copies, with most showing recent insertions (K divergence ≈ 0), indicating ongoing transpositional activity. Notably, PB elements from *Harmonia axyridis*, *Apoderus coryli*, and *Diabrotica balteata* exhibit exceptional potential for genetic tool development. Structurally, intact PB elements ranged from 2074 to 3465 bp, each containing a single transposase ORF (500–725 aa). All were flanked by terminal inverted repeats and generated TTAA target site duplications. **Conclusions**: These findings demonstrate PB transposons have not only shaped historical beetle genome evolution but continue to drive genomic diversification, underscoring their dual significance as natural genome architects and promising biotechnological tools.

## 1. Introduction

The piggyBac (PB) transposon was first identified in the cabbage looper moth (*Trichoplusia ni*) as an insertional mutagen during baculovirus infections [1]. Subsequent studies revealed that the PB superfamily is widely distributed across animals, fungi, and protists, with occasional occurrences in plant genomes (e.g., red algae) [2,3]. While PB evolution in animals has been extensively documented, studies in fungi and plants remain limited. Horizontal transfer likely facilitated its widespread dissemination across species, and PB transposons are naturally present in diverse animal genomes, ranging from insects to mammals [4,5,6,7,8,9,10], including primates [11]. Notably, a naturally active piggyBac transposon was identified in bats, named as piggyBat [12]. Extensive domestication of PB-derived sequences has been observed in vertebrates, with five domesticated piggyBac genes identified in humans. These genes have lost their transposition activity but have acquired novel cellular functions [2,13].

Due to its high transposition efficiency, precise “cut-and-paste” mechanism, and broad host range, the PB transposon has been engineered as a powerful tool for genetic manipulation [14,15,16,17]. It has been successfully employed in model organisms such as mice, rats, and non-human primates [18,19,20,21]. PB transposons enable insertional mutagenesis by integrating into genes or regulatory regions, providing a valuable platform for functional genomics [22,23,24,25]. Additionally, they serve as a precise marker excision system in plants [26,27], algae [28], fungi [29], and mammals [30], facilitating targeted genetic modifications. A key advantage of PB transposons is their utility as a non-viral alternative for gene therapy, offering a simpler, safer, and more versatile approach compared to viral vectors [31].

Beetles (Coleoptera) represent one of the most diverse and ecologically significant animal groups, comprising over 400,000 described species—nearly 25% of all known animal life [32]. Their evolutionary success and adaptability make them a focal point for biological research. PB transposons, which may contribute to genomic evolution in beetles, have been identified in Tenebrionidae species, and studies reveal a diverse array of PB-like elements in the *Tribolium castaneum* genome [33,34], suggesting active transposition in this lineage. However, the broader evolutionary profile of PB transposons in beetles—including their distribution, activity patterns, and functional impact—remains poorly understood.

In this study, we systematically investigate the evolutionary landscape of PB transposons in beetles, including their distribution, abundance, structural organization, and evolutionary dynamics. Accurate annotation and classification of transposons in genome assemblies present known challenges, such as their repetitive nature and sequence diversity. Our methodology, employing a combined approach of homology searches and structural analysis, is designed to overcome these limitations and provide a comprehensive profile. These insights may elucidate their role in shaping genetic diversity and adaptation in beetles. Furthermore, studying PB dynamics in beetles enhances our understanding of transposon–host coevolution in insects. Evolutionary analyses and activity predictions of beetle-specific PB systems may also facilitate the discovery of active PB elements and the development of novel genetic tools.

## 2. Materials and Methods

### 2.1. PB Transposon Miningin Coleoptera

Among all the nearly 600 coleoptera genomes in the NCBI, some were duplicate entries for the same species, while others had poor assembly quality. The genomes we selected underwent a systematic filtering process, where only one genome per species was retained, and only those meeting specific assembly quality criteria were included. Thus, a total of 136 coleopteran insect genomes (high assembly quality) were retrieved and downloaded from the NCBI database (https://www.ncbi.nlm.nih.gov/datasets/genome/, accessed on 5 December 2024). For each obtained genome, a local tBLASTN search was performed using the DDE domain of the PB transposase as the query. Sequences exhibiting > 30% coverage and >80% identity to the query were retained, along with their flanking genomic regions extended by 4 kb upstream and downstream. These sequences were clustered using USEARCH software (v11.0.667) with an identity threshold of 0.5. Clustered sequences were subsequently aligned at their boundaries using the MAFFT plugin implemented in BioEdit (v7.2.6). Manual inspection was conducted to verify structural features including terminal inverted repeats (TIRs), target site duplications (TSDs), and transposase domains. Consensus or representative sequences were extracted from validated clusters.

### 2.2. Phylogenetic Tree Construction

Phylogenetic analysis was performed on 19 PB transposase reference sequences from previous reports and 17 IS1380 transposases downloaded from ISfinder (https://www-is.biotoul.fr/index.php, accessed on 5 December 2024), combined with newly identified PB transposases from Coleoptera. Multiple sequence alignment was conducted using the G-INS-I algorithm in MAFFT (v7.471) [35]. The maximum likelihood phylogenetic tree was subsequently constructed with IQ-TREE (v1.6.1) employing an ultrafast bootstrap approach with 1000 replicates, using IS1380 transposases as the outgroup [36]. For transposable element TIR conservation analysis, representative sequences from each phylogenetic branch were selected. 20 bp TIR regions were extracted and subjected to consensus motif visualization using WebLogo 3 (https://weblogo.berkeley.edu/logo.cgi, accessed on 14 April 2025).

### 2.3. Evolutionary Dynamics Analysis

To compare the dynamics of diverse PB elements across species, representative PB elements were selected from various species based on criteria including the presence of characteristic TSD and TIR, encoding of a complete transposase, and having more than 10 intact copies (PB transposon encoding intact transposase, flanked by left and right TIRs). Consensus or representative sequences were extracted using BioEdit (v7.2.6). Kimura divergence (K) was calculated using the calcDivergenceFromAlign.pl script from the RepeatMasker package [37]. Data visualization was performed using GraphPad Prism (v8.0.2).

### 2.4. PB Sequence Analysis

Potential open reading frames (ORFs) were identified through computational prediction using GENSCAN (http://hollywood.mit.edu/GENSCAN.html, accessed on 10 April 2025). Subsequent protein domain characterization was performed through profile hidden Markov model (pHMM) analysis implemented via the hmmscan web server (EMBL-EBI; https://www.ebi.ac.uk/Tools/hmmer/search/hmmscan, accessed on 12 April 2025).Structural visualization of PB transposase was generated using Illustrator for Biological Sequences (IBS v1.0.3) [38]. Sequence alignments were conducted with MAFFT (v7.471) and rendered using Jalview (v2.11.4.0). Comparative sequence analysis focused on conserved domains including terminal inverted repeats (TIRs), DDE domains (DDD), and DNA-binding domains (DDBD1), with pairwise identity calculations performed in BioEdit (v7.2.6). Resultant identity matrices were transformed into heatmap visualizations using GraphPad Prism (v8.0.2). Sequence conservation patterns were quantitatively characterized using WebLogo 3 (https://weblogo.berkeley.edu/logo.cgi, accessed on 14 April 2025) with Shannon entropy normalization.

## 3. Results

### 3.1. Distribution

Data mining revealed that PB transposons are widely distributed across beetles (Coleoptera). We identified 645 PB transposon copies (including 534 intact copies of PB transposons) from 136 beetle species (representing 62 genera from 11 families), with 195 PB transposons detected in 62 species spanning 59 genera and 11 families (Figure 1, Table 1). The copy number of full-length, intact PB transposons varied significantly among species, ranging from <10 copies in many to >50 copies in some. Additionally, multiple independent PB invasions were observed across most species, resulting in diverse PB transposon lineages (as illustrated in Figure 2a). Notably, we detected more than 20 distinct PB elements in *Holotrichia oblita*, *Acanthoscelides obtectus*, *Agriotes lineatus*, and *Chrysolina oricalcia* genomes. The highest PB transposon diversity was found in *Diabrotica balteata*, which harbored 67 PB elements-the largest repertoire identified in this study (Appendix A).

### 3.2. Classification and Structure Organization

Phylogenetic analysis classified PB transposases into six distinct clades with high bootstrap supports (≥75%) (Figure 2a). The structural organization of PB elements was illustrated using representative members from each clade, as summarized in Appendix A and Figure 2b. Intact PB elements varied in length from 2074 bp to 3465 bp, each containing a single open reading frame (ORF) that encodes a transposase (~500–725 aa). All PB elements were flanked by short terminal inverted repeats (TIRs; ≤26 bp) and TTAA target site duplications (Appendix A and Figure 2b).

Notably, an analysis of the generalized TIR sequence logo (24 bp) revealed that TIRs exhibit considerable divergence across the six clades (Figure 2b). However, the terminal ends of nearly all TIRs were highly conserved, as exemplified by the 5′ end, which showed a pronounced enrichment of C and T nucleotides. Six well-conserved motifs were identified at the 5′ ends of TIRs: CCCT, CCCTTT, CACGTT, CCCTTT, CCCATT, and CCCTTT, with CCCT being the most prevalent (Figure 2c). Additionally, most PB transposon elements consistently displayed a TTAA target site duplication (TSD), and a small number of detected elements either lack identifiable TSDs or show deviations from the canonical TTAA site (Figure 2b and Appendix A).

### 3.3. Domain Organization and Sequence Conservation of PB Transposases

As previously reported [8], PB transposases contain five conserved domains (Figure 3a): the N-terminal domain (NTD), dimerization and DNA-binding domain 1 (DDBD1), catalytic domain (DDD/DDE), dimerization and DNA-binding domain 2 (DDBD2), and the C-terminal cysteine-rich domain (CRD). Comparative analysis revealed that full-length PB transposases display considerable diversity, with 7.1–10.6% sequence identity within clades and 6.5–7.6% between clades. The catalytic DDD domain was the most conserved, exhibiting 43–56% identity within clades and 34–43% between clades (Figure 3b). Notably, the three critical catalytic residues (DDD) were highly conserved across all six clades (Figure 3b). In contrast, other domains showed limited conservation: NTD: extremely divergent (≤12% identity between clades, DDBD1: 29–33% identity, DDBD2: 22–28% identity, and CRD: 22–37% identity (Figure 3b).

### 3.4. Evolutionary Dynamics of PB Transposons in the Genomes of Beetles

The evolutionary dynamics (or activity status) of a specific transposon in a genome can be well reflected by two key characteristics: Kimura divergence and intact copies. Intact copies refer to transposons that encode an intact transposase and are flanked by functional terminal inverted repeats (TIRs); these are considered potentially active for transposition. Kimura divergence is a measure of genetic divergence between transposon copies that accounts for different rates of nucleotide substitution [39]. It reflects the level of mutation accumulation and, on a relative time scale, the activity of a transposon within a genome. Older, long-invaded transposons tend to accumulate more mutations, resulting in higher Kimura divergence, whereas younger or recently invaded transposons tend to accumulate fewer mutations, show lower Kimura divergence, and are more likely to be transpositionally active. This hypothesis has been evaluated in previous studies through mining and testing of ZB [40], PS [41], Buster [42] and Spy [43] transposons.

Our data mining revealed that intact PB copies were identified in all PB-detected species, suggesting that some PB transposon copies represent recent invasions and may reflect ongoing or current activity in these genomes. The Kimura divergence of PB transposons were estimated for some species containing putatively active PB (intact copies > 50% and total intact copies > 10) using Kimura divergence (K) values [39] as described in Methods. Our analysis demonstrated that most PB transposons exhibit very young insertion ages (K ≈ 0), suggesting recent genomic invasions and potential transposition activity (Figure 4 and Appendix A). Interestingly, both young and old PB transposons coexist in several species, including *Diabrotica balteata* (Diba), *Apoderus coryli* (Apco), indicating recurrent PB transposon invasions in these genomes. Some genomes appear to represent long-term and persistent invasions of PB transposons (Figure 4), such as PB-HoOB-3, PB-HoOB-15, PB-HoOB-28 in *Holotrichia oblita* (Hoob), PB-LoCr-4 in *Lochmaea crataegi* (Locr), PB-DiBa-5, PB-DiBa-26 in *Diabrotica balteata* (Diba). Notably, PB transposons appear to be currently active in certain species, particularly those containing elements with both extremely young insertion ages (K ≈ 0) and 100% intact copies. Prominent examples include: PB-HaAx-2 in *Harmonia axyridis* (Haax), PB-ApCo-6 in *Apoderus coryli* (Apco), and PB-DiBa-3 in *Diabrotica balteata* (Diba) (Figure 4 and Appendix A). These elements represent excellent candidates for developing new gene transfer tools due to their recent activity and high integrity.

## 4. Discussion

Large variations in genome size—ranging from 154 to 2578 megabases—have been documented in beetles [44,45]. A widely held view is that transposable elements (TEs), including both DNA transposons and retrotransposons, are a major contributor to this genome size variation [46,47,48]. Furthermore, chromosomal rearrangements and the reorganization of satellite DNAs derived from TEs appear to be linked, with extensive genomic reshuffling observed in beetle species exhibiting reduced diploid numbers. Supporting data also reinforce the role of TEs in the origin of satellite DNAs, which can spread widely across the genome, including euchromatic regions [49].

Case studies illustrate the dynamic impact of TEs. For instance, the agricultural pest beetle Sitophilus oryzae has experienced successive bursts of TE amplification, with TEs now comprising 72% of its genome [50]. Similarly, in the Colorado potato beetle (*Leptinotarsa decemlineata*), TE insertion sites differ between geographic populations; overall TE diversity is sufficient to distinguish populations, and more TEs are found in beetles from Mexico than in those from the United States, reflecting differential evolutionary histories [51]. These findings provide new insights into the evolutionary dynamics of TEs and underscore their profound impact on genome architecture and evolution.

Nevertheless, the evolutionary landscapes of DNA transposons and retrotransposons in beetles remain poorly characterized. In this study, we systematically investigated the evolutionary landscape of PB transposons, a superfamily of DNA transposons known for their high activity in insects [6,7,52]. We focused on beetles (Coleoptera)—the most species-rich animal order and a highly diverse arthropod group—revealing the wide distribution, high diversity, and notable activity of PB transposons in this lineage.

Our analysis revealed an uneven distribution of PB elements across beetles: they were detected in nearly half (62 out of 136) of the species examined, while absent in the other half. Moreover, PB transposases displayed substantial diversity, with six distinct clades identified. Coupled with highly variable copy numbers—ranging from fewer than 10 copies in many species to over 50 in some—these patterns reflect differential evolutionary histories of PB elements among beetle lineages.

Importantly, we found a high number of intact PB copies across species, and many harbored recently invaded elements with Kimura divergence (K) values close to zero, indicating sustained transpositional activity. These results establish PB transposons as genomic parasites that have significantly influenced historical genome evolution in beetles and are likely continuing to drive ongoing genomic diversification. Further investigation of PB insertion polymorphisms among individuals and populations would help confirm this hypothesis.

Due to their recent activity and high integration efficiency, PB transposons are excellent candidates for developing novel genetic tools. Notably, PB can be engineered for precise excision without leaving a genomic footprint [40], unlike most Tc1-family transposons (e.g., SB [41] and ZB [42]), which typically leave 4–6 bp remnants. To date, five transcriptionally active PB transposons have been identified, four of which originate from arthropods: the cabbage looper moth (*Trichoplusia ni*) [1], fall armyworm (*Spodoptera frugiperda*) [43], rice striped stem borer (*Chilo suppressalis*) [44], and brown planthopper (*Nilaparvata lugens*) [7]. At least three have been successfully developed into high-efficiency genetic tools: the T. ni-derived PB for gene transfer [17] and footprint-free excision [45]; Magease (MG) from *S. frugiperda* [43]; and the naturally active piggyBat from bats, which has been optimized for high efficiency [46]. PB elements from *C. suppressalis* and *N. lugens* remain transcriptionally active but show relatively low transposition efficiency in their unmodified forms.

In this study, we observed widespread, recent, and ongoing PB transposon activity in beetle genomes. Several species harbor PB variants with exceptionally high proportions of intact copies and very young insertion ages—indicated by Kimura divergence (K) values approaching zero—such as PB-HaAx-2 in *Harmonia axyridis*. Characterized by structural integrity and recent transposition activity, in contrast to ancient, fragmented PB copies, these elements represent promising candidates for next-generation genetic tool development and merit further functional evaluation. Testing the activity of these candidates in mammalian cells would be a valuable next step to confirm their functional utility and broader applicability.

## Figures and Tables

**Figure 1 genes-16-01521-f001:**
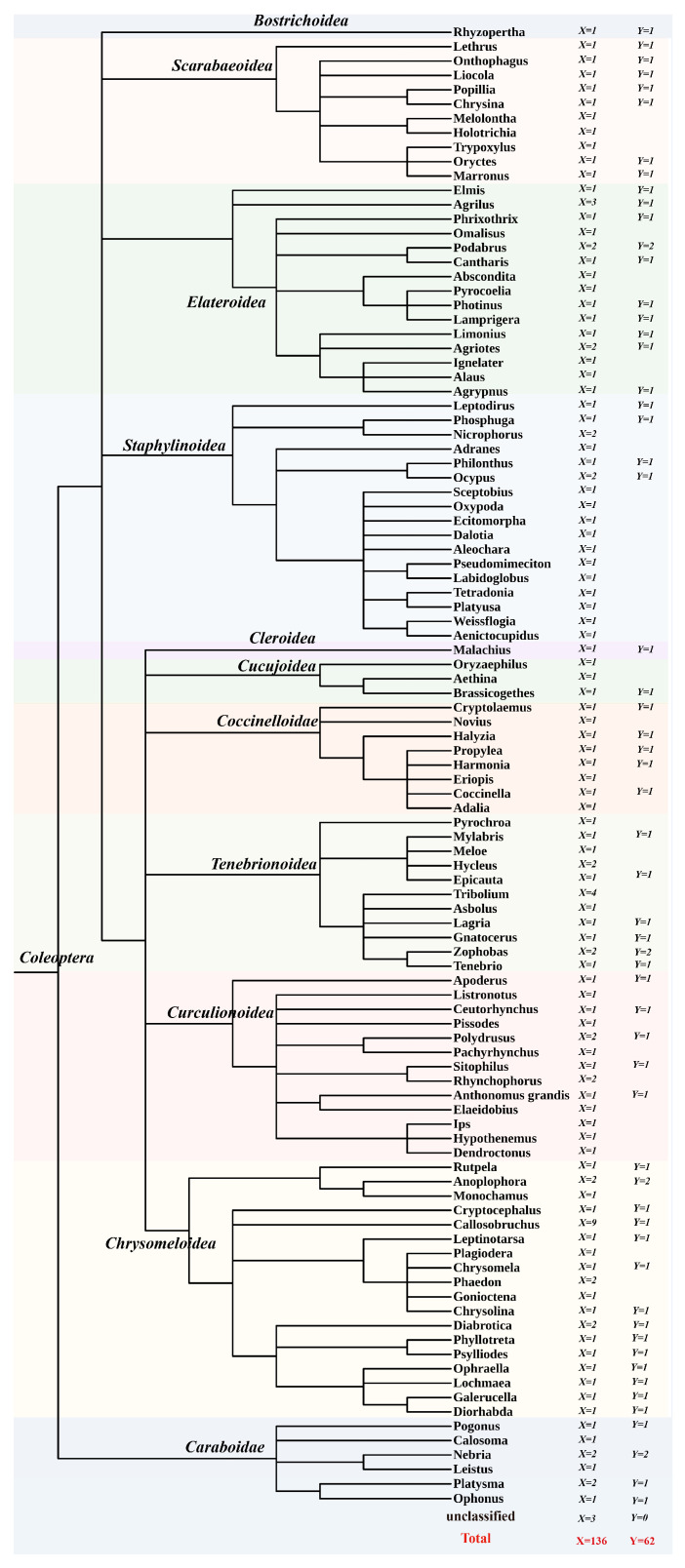
Taxonomic Distribution of PiggyBac (PB) Elements in Beetles. Species distribution map constructed from phylogenetic trees of each species, with trees retrieved from NCBI (Taxonomy browser). The tree root is Coleoptera (Order); each leaf of the phylogenetic tree represents a genus, with superfamily names labeled on branches. Each leaf denotes a genus. The X indicates the number of species assayed per genus, and the Y represents the number of those species in which the PiggyBac transposon was detected. Different colors are used for better distinction among superfamilies.

**Figure 2 genes-16-01521-f002:**
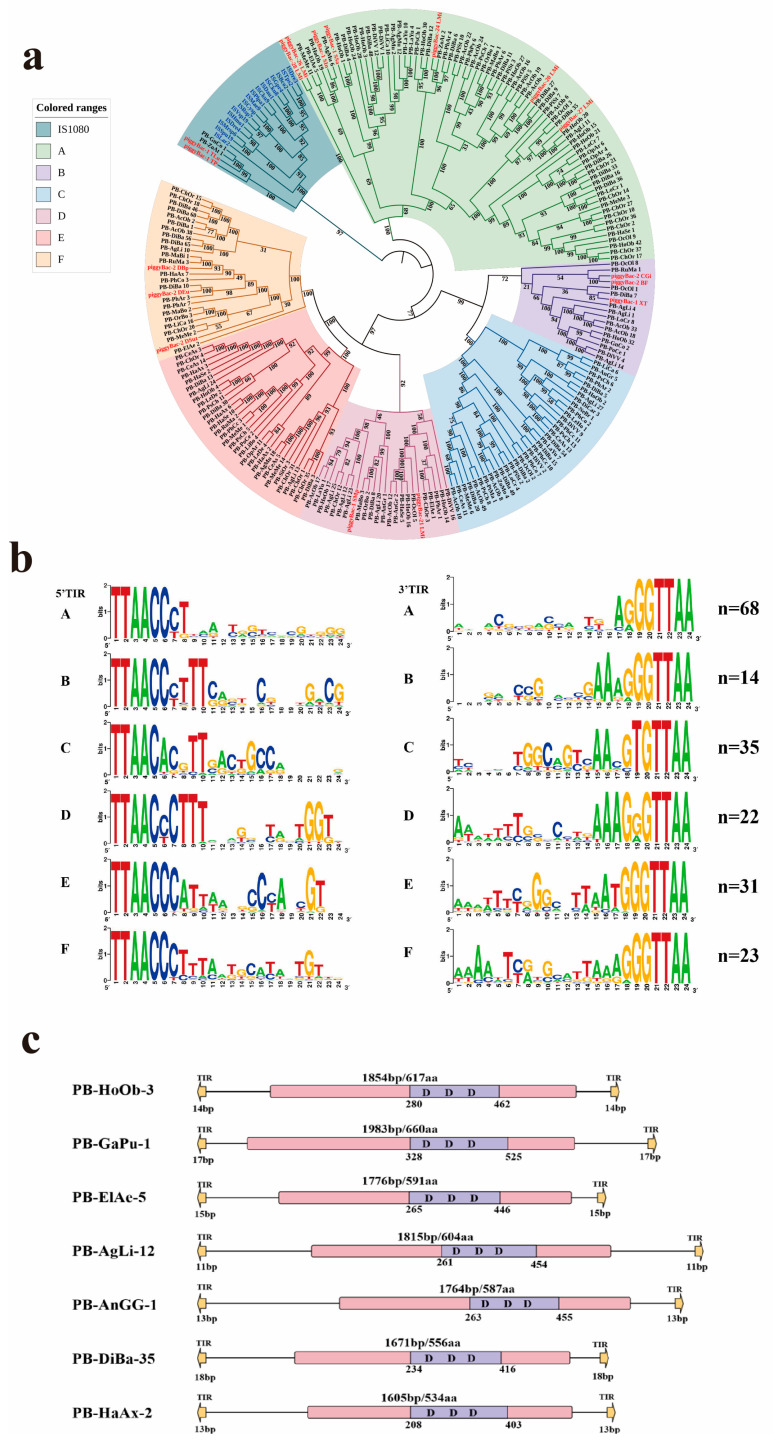
Classification and structural organization of piggyBac (PB) transposons. (**a**) Phylogenetic tree constructed from PB transposons identified in animals in this study and 34 reference sequences (using *IS1380* as the outgroup). All PB elements were classified into six distinct clades (A–F). Detected transposons are labeled in black using abbreviations formed from the first two letters of each word in the species name, with different sequences from the same species distinguished by numerical identifiers. Reference sequences used as the outgroup are labeled in blue, and reference sequences for PiggyBac transposons are labeled in red. (**b**) Sequence logo illustrating the conserved terminal inverted repeats (TIRs) of PB elements. The first 24 bp of TIR sequences from both ends were analyzed using WebLogo 3 (WebLogo-Create Sequence Logos), where nucleotide height reflects sequence conservation. the n value in the upper right corner represents the number of sequences used for calculation and logo generation. (**c**) Schematic representation of PB transposon architectures. Key structural features are annotated: yellow arrows denote TIR regions, purple boxes represent conserved DDD catalytic domains, and pink boxes indicate transposase-coding regions.

**Figure 3 genes-16-01521-f003:**
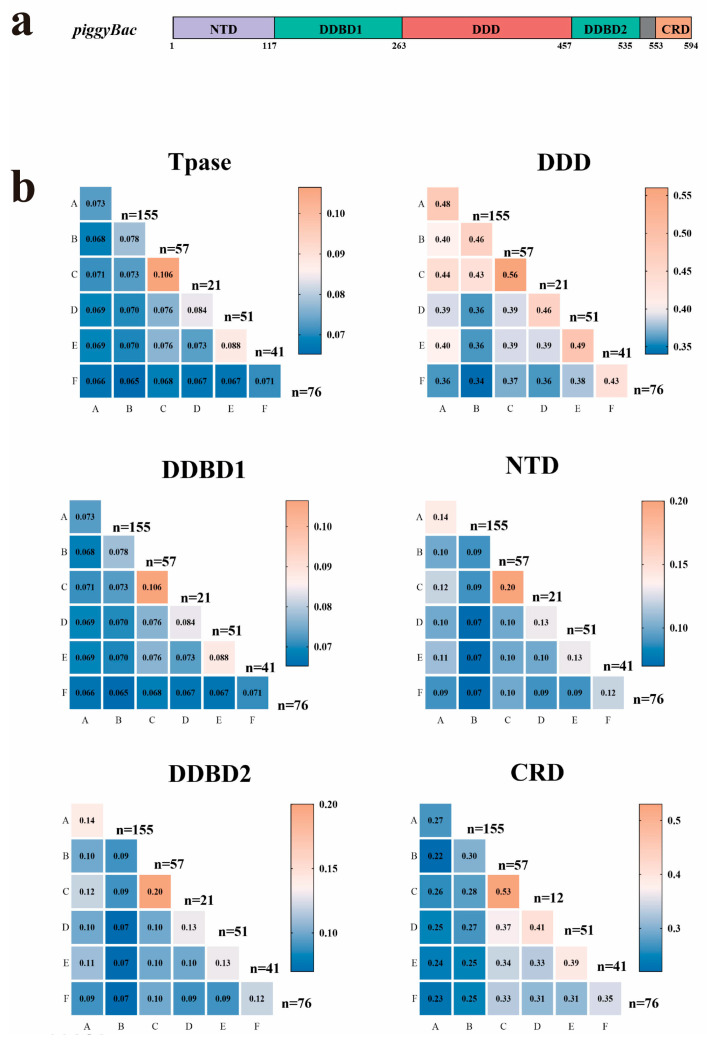
Heatmap visualization of sequence conservation across transposon clades A–F. (**a**) Schematic representation of piggyBac (PB) transposase domains. (**b**) Numerical values indicate mean percentage identity derived from pairwise comparisons of full length of transposase (Tpase) and distinct structural domains. Catalytic domain (DDD), N-terminal domain (NTD), Dimerization and DNA binding domain 1 (DDBD1), Dimerization and DNA binding domain 2 (DDBD2), and C-terminal cysteine-rich domain (CRD). Values were computed through comprehensive alignment of corresponding domain sequences between clade pairs. The numerical values in the heatmap represent the sequence similarity of domains within a single clade or between two different clades. Lower similarity is shown in blue, while higher similarity is shown in orange. The value of n indicates the number of sequences from each clade used in the calculation.

**Figure 4 genes-16-01521-f004:**
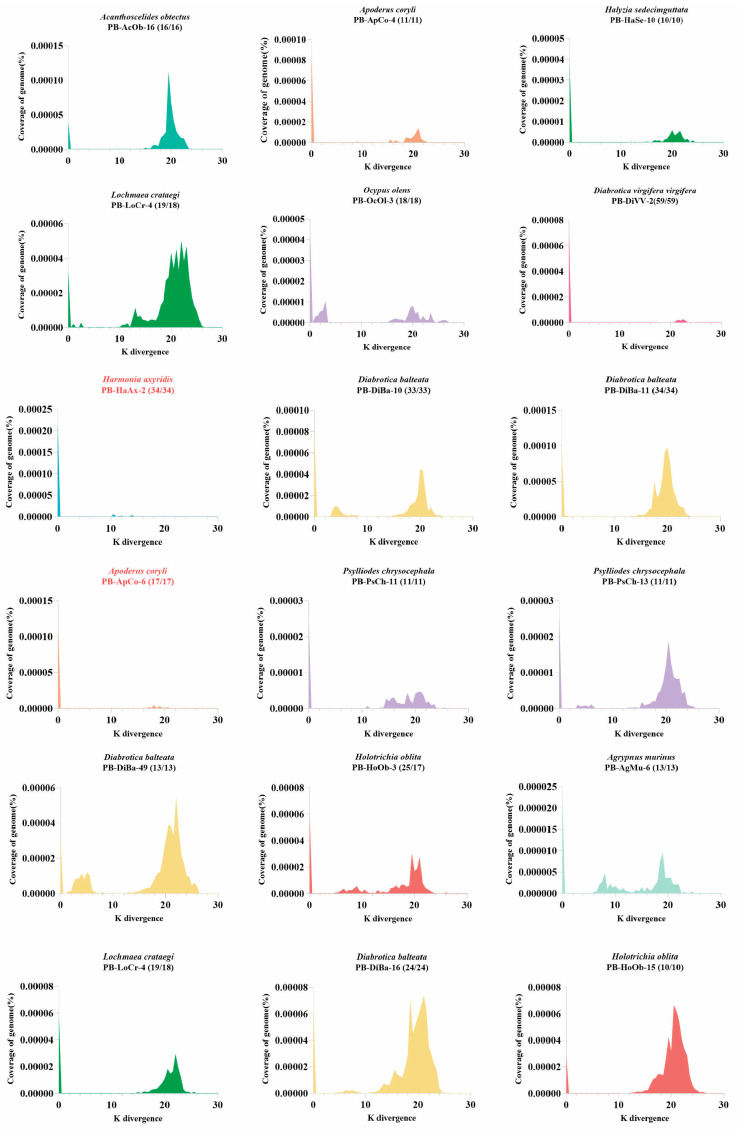
Evolutionary dynamics of 18 piggyBac (PB) transposons in Coleoptera. The transposon insertion ages were estimated based on K divergences as described in methods. Temporal dynamics are visualized as K divergence (X-axis) versus genomic coverage (% of total genome; Y-axis). Bracketed values indicate intact copies/total PB copies per genome. Different species are represented by distinct colors. Transposons more likely to be active are highlighted in red. 18 PB transposons were selected based on sequence identity of PB copies in genome (>99%). Appendix A provides the set of PB transposase sequences used in the analysis, ordered by decreasing identity.

**Table 1 genes-16-01521-t001:** Characterization of PB transposable elements in Coleoptera.

Distribution	Number of Species	Number of PBs ^a^	Number of Intact PBs ^b^	Length of Intact PB	Transposase Length of Intact PB	TIR Length of Intact PB (bp)
*Carabidae*	5	25	20	2261–2884	501–644	12–28
*Chrysomelidae*	18	264	221	2078–3214	502–725	7–31
*Melyridae*	1	9	8	2404–2884	589–692	12–19
*Coccinellidae*	4	30	28	2304–2637	512–640	8–25
*Nitidulidae*	1	8	7	2277–2983	587–649	10–19
*Attelabidae*	1	6	6	2403–2757	508–668	13–19
*Curculionidae*	6	64	45	2271–3083	506–648	9–30
*Elateridae*	11	113	90	2087–3083	505–668	8–34
*Scarabaeidae*	5	80	68	2242–3199	514–655	6–46
*Staphylinidae*	4	24	23	2074–3465	502–711	12–26
*Tenebrionidae*	6	22	18	2110–3008	500–627	9–20
Total	62	654	534			

^a^ PB: Transposable elements bounded by identifiable terminal inverted repeats (TIRs) and flanking target site duplications (TSDs). ^b^ intact PB: Functional transposons containing both diagnostic TIRs and TSDs, with complete coding capacity for transposase proteins exceeding 500 amino acids in length.

## Data Availability

All newly identified PB transposon sequences reported in this study have been deposited in the NCBI database. The corresponding accession numbers for each sequence are listed in Appendix A and can be accessed directly via NCBI.

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
