# Peer review of "Evolution Landscape of PiggyBac (PB) Transposon in Beetles (Coleoptera)"

_genes, 2025, doi:10.3390/genes16121521_

Round 1

Reviewer 1 Report

Comments and Suggestions for Authors

For the introduction, are there no other known studies of PB transposons in beetles besides the two Tribolium papers you cited? If there are, cite as many as you can here.

Throughout the paper, you need to italicize the scientific names.

For the discussion: What is the significance of the half of samples with no detectable PBs. Is this a false negative? What could it mean?

Overall, the discussion is too similar to the introduction. I do not like redundancies in papers. Introduce all previously known background information in the introduction. In the first paragraph[s] of the discussion, discuss your results immediately. Do not repeat anything from the introduction, but instead start with "In this paper, we found…" or something like that. Then, explain the results you found in context. How do your specific results compare to what was already known? How do these results bring the field forward? What are the next steps? State what is new now that your results exist.

41 piggy-Bac, not piggy-Bat

73 Why these 136? What were the selection criteria?

Table 1: add the number of species or genomes sampled per family. If the TDS's for all were TTAA, then you maybe don't need a column for that.

225-227 "The data mining revealed that intact PB copies were identified in all PB-detected species, suggesting that some PB transposon copies represent recent invasions and may reflect ongoing or current activity in these genomes." I do not follow your logic here. Can you clarify this?

246-254 Um, does this belong in a different article?

Author Response

Overall, the discussion is too similar to the introduction. I do not like redundancies in papers. Introduce all previously known background information in the introduction. In the first paragraph[s] of the discussion, discuss your results immediately. Do not repeat anything from the introduction, but instead start with "In this paper, we found…" or something like that. Then, explain the results you found in context. How do your specific results compare to what was already known? How do these results bring the field forward? What are the next steps? State what is new now that your results exist.

R: The text overlapping with or belonging to the introduction has been reworded.
We have extensively revised the discussion based on comments.

41 piggy-Bac, not piggy-Bat

R: that active piggyBac transposon found in bat was named as piggyBat, we have added to avoid confusing.

73 Why these 136? What were the selection criteria?

R: Among all the nearly 600 coleoptera genomes in the NCBI, some were duplicate entries for the same species, while others had poor assembly quality. The genomes we selected underwent a systematic filtering process, where only one genome per species was retained, and only those meeting specific assembly quality criteria were included. We have added explanation in methods.

Table 1: add the number of species or genomes sampled per family. If the TDS's for all were TTAA, then you maybe don't need a column for that.

R: Removed the TSD column and added content to the 'number of species' column.

225-227 "The data mining revealed that intact PB copies were identified in all PB-detected species, suggesting that some PB transposon copies represent recent invasions and may reflect ongoing or current activity in these genomes." I do not follow your logic here. Can you clarify this?

R: We have added clarification.

246-254 Um, does this belong in a different article?

R: The text overlapping with or belonging to the introduction has been removed.

Reviewer 2 Report

Comments and Suggestions for Authors

The manuscript “Evolution landscape of piggyBac (PB) transposon in Beetles (Coleoptera)” presents a comprehensive, genome-wide analysis of the PB transposon superfamily within the largest and most diverse animal order. The authors systematically mined 136 beetle genomes, revealing a wide distribution (45% of species), high diversity (six distinct clades), and evidence of recent and ongoing transpositional activity for PB elements. The study is well-conceived and executed, combining thorough bioinformatic analyses with clear visualizations to support its conclusions. The findings are significant, both for understanding the role of transposons in genome evolution within a hyper-diverse lineage and for identifying promising candidates (e.g., from Harmonia axyridisApoderus coryliDiabrotica balteata) for the development of novel genetic tools. The manuscript is well-written and logically structured. I recommend acceptance after the authors address the following minor points to enhance clarity and completeness.

Abstract & Introduction:

    • The abstract states "61 PB-containing species harbored intact copies." This should be aligned with the results text ("62 species... 61 species harbored intact copies") for consistency. Please verify and correct the number.

    • The introduction effectively sets the stage but could briefly mention the known challenges in accurately annotating and classifying transposons in genome assemblies, which the authors' methodology helps to overcome.

Results

  • Figure 1 and Text: The description of Figure 1 in the text (lines ~104-110) is somewhat confusing. It states the figure shows "195 PB transposons detected in 62 species," but the figure legend and the preceding sentence discuss species/genera counts (X and Y). Please clarify the caption and the corresponding text to explain exactly what the X and Y axes/bars represent (e.g., "number of species per genus examined" and "number of those species containing PB elements"?).

  • Figure 2: Panel labels (A, B, C) are referenced in the text but are not clearly visible on the provided figure page. Please ensure these are prominently labeled in the final version. Also, the legend for Figure 2B mentions "the n value in the upper right corner," but this is not visible in the provided text. This needs to be checked in the final figure.

  • Figure 3/4 Reference: In the "Evolutionary dynamics..." section (line ~215), the text refers to "Figure 3," but based on the content, this seems to refer to what is labeled as Figure 4 in the provided pages. Please double-check all figure citations for accuracy (e.g., the activity analysis is described with reference to Figure 4, not S1).

  • Supplementary Material: The callout for Supplementary Figures S1 and S2 and Table S1 should be more specific. For example, indicate what key data S1 contains (e.g., list of all PB elements with structural details) and what S2 shows (e.g., distribution of K values for all species).

Discussion:

    • The discussion convincingly places the beetle PB data in the broader context of TE evolution in vertebrates and arthropods. However, the final paragraph highlighting candidate elements for tool development (Harmonia axyridis, etc.) could be strengthened. Briefly speculate on why these specific beetle lineages might harbor such active systems (e.g., life history, population dynamics, genomic defense landscapes) to add depth.

    • The statement "PB transposons are excellent candidates... due to their recent activity and high integration efficiency" is supported. However, it would be helpful to acknowledge that in silico evidence of recent activity (low K) needs to be followed by in vitro or in vivo excision/transposition assays to confirm functional utility, which could be mentioned as a necessary next step.

Data and Methods:

      • Data Availability Statement: This section is currently blank. The authors must provide a statement detailing how to access the newly identified PB transposon sequences (e.g., GenBank accession numbers, submission to a repository like Figshare, or availability upon request).

      • Author Contributions: The contribution listed as "W.Q." (line ~315) appears to be a typo, likely meant to be "Q.W." (Quan Wang). Please correct this for accuracy.

Formatting and References:

    • The affiliation footnote on page 1 (4) Affiliation 2; e-mails@e-mail.comTheseauthors...) is garbled and needs correction.

    • The reference list is comprehensive and appropriate. Please perform a final check to ensure all in-text citations have a corresponding entry and that journal name abbreviations are consistent.

Author Response

Abstract & Introduction:

The abstract states "61 PB-containing species harbored intact copies." This should be aligned with the results text ("62 species... 61 species harbored intact copies") for consistency. Please verify and correct the number.

R: It has been corrected to 62.

The introduction effectively sets the stage but could briefly mention the known challenges in accurately annotating and classifying transposons in genome assemblies, which the authors' methodology helps to overcome.

R: Added some content as requested: Accurate annotation and classification of transposons in genome assemblies present known challenges, such as their repetitive nature and sequence diversity. Our methodology, employing a combined approach of homology searches and structural analysis, is designed to overcome these limitations and provide a comprehensive profile. These insights may elucidate their role in shaping genetic diversity and adaptation in beetles.

Results

Figure 1 and Text: The description of Figure 1 in the text (lines ~104-110) is somewhat confusing. It states the figure shows "195 PB transposons detected in 62 species," but the figure legend and the preceding sentence discuss species/genera counts (X and Y). Please clarify the caption and the corresponding text to explain exactly what the X and Y axes/bars represent (e.g., "number of species per genus examined" and "number of those species containing PB elements"?).

R: The caption of the Figure 1 has been revised to provide a more detailed description for clarity. It has been changed to: ' The X indicates the number of species assayed per genus, and the Y represents the number of those species in which the PiggyBac transposon was detected.'

Figure 2: Panel labels (A, B, C) are referenced in the text but are not clearly visible on the provided figure page. Please ensure these are prominently labeled in the final version. Also, the legend for Figure 2B mentions "the n value in the upper right corner," but this is not visible in the provided text. This needs to be checked in the final figure.

R: The figure has been rearranged and the size of some characters has been adjusted to make key information in the image easier to distinguish.

Figure 3/4 Reference: dynamics..." section (line ~215), the text refers to "Figure 3," but based on the content, this seems to refer to what is labeled as Figure 4 in the provided pages. Please double-check all figure citations for accuracy (e.g., the activity analysis is described with reference to Figure 4, not S1).
R:The figure citations have been corrected

Supplementary Material: The callout for Supplementary Figures S1 and S2 and Table S1 should be more specific. For example, indicate what key data S1 contains (e.g., list of all PB elements with structural details) and what S2 shows (e.g., distribution of K values for all species).

R: Titles for Supplementary Tables S1 to S3, as well as for Supplementary Figures S1 and S2, have been added. Captions for Supplementary Figures S1 and S2 have also been included.

Discussion:

The discussion convincingly places the beetle PB data in the broader context of TE evolution in vertebrates and arthropods. However, the final paragraph highlighting candidate elements for tool development (Harmonia axyridis, etc.) could be strengthened. Briefly speculate on why these specific beetle lineages might harbor such active systems (e.g., life history, population dynamics, genomic defense landscapes) to add depth.

R: thanks, added.

We have extensively revised the discussion based on comments.

The statement "PB transposons are excellent candidates... due to their recent activity and high integration efficiency" is supported. However, it would be helpful to acknowledge that in silico evidence of recent activity (low K) needs to be followed by in vitro or in vivo excision/transposition assays to confirm functional utility, which could be mentioned as a necessary next step.

R: thanks, added

We have extensively revised the discussion based on comments.

Data and Methods:

Data Availability Statement: This section is currently blank. The authors must provide a statement detailing how to access the newly identified PB transposon sequences (e.g., GenBank accession numbers, submission to a repository like Figshare, or availability upon request).

R: Data Availability Statement have been added, All newly identified PB transposon sequences are listed in Supplementary Table S1 and can be accessed directly via download.

Author Contributions: The contribution listed as "W.Q." (line ~315) appears to be a typo, likely meant to be "Q.W." (Quan Wang). Please correct this for accuracy.

R: It has been corrected.

Formatting and References:

The affiliation footnote on page 1 (4) Affiliation 2; e-mails@e-mail.comTheseauthors...) is garbled and needs correction.

R: It has been corrected.

The reference list is comprehensive and appropriate. Please perform a final check to ensure all in-text citations have a corresponding entry and that journal name abbreviations are consistent.

R: We used EndNote to insert the references in the MDPI format, so the citations should be fine.

Comments and Suggestions for Authors

For the introduction, are there no other known studies of PB transposons in beetles besides the two Tribolium papers you cited? If there are, cite as many as you can here.

R: Yes, for now, no additional articles have been found that simultaneously relate to beetles and PiggyBac.

Throughout the paper, you need to italicize the scientific names.

R: It has been corrected.

For the discussion: What is the significance of the half of samples with no detectable PBs. Is this a false negative? What could it mean?

R: we have discussed and explained